# Comparative Effectiveness of Western and Eastern Manual Therapies for Chronic Obstructive Pulmonary Disease: A Systematic Review and Network Meta-Analysis

**DOI:** 10.3390/healthcare9091127

**Published:** 2021-08-30

**Authors:** Chan-Young Kwon, Boram Lee, Beom-Joon Lee, Kwan-Il Kim, Hee-Jae Jung

**Affiliations:** 1Department of Oriental Neuropsychiatry, Dong-eui University College of Korean Medicine, Busan 47227, Korea; beanalogue@naver.com; 2Clinical Research Coordinating Team, Korea Institute of Oriental Medicine, Seoul 02447, Korea; qhfka9357@naver.com; 3Department of Internal Korean Medicine, Kyung Hee University Korean Medicine Hospital, Seoul 02453, Korea; franchisjun@naver.com; 4Division of Allergy, Immune and Respiratory System, Department of Internal Medicine, College of Korean Medicine, Kyung Hee University, Seoul 02447, Korea

**Keywords:** chronic obstructive pulmonary disease, manual therapy, systematic review, meta-analysis, network meta-analysis

## Abstract

Background: Manual therapy (MT) is considered a promising adjuvant therapy for chronic obstructive pulmonary disease (COPD). Comparing the effectiveness among different Western and Eastern MTs being used for the management of COPD could potentially facilitate individualized management of COPD. This systematic review attempted to estimate the comparative effectiveness of Western and Eastern MTs for COPD patients using a network meta-analysis (NMA) methodology. Methods: Nine electronic databases were comprehensively searched for relevant randomized controlled trials (RCTs) published up to February 2021. Pair-wise meta-analysis and NMA were conducted on the outcomes of COPD, which included lung function and exercise capacity. Results: The NMA results from 30 included RCTs indicated that the optimal treatment for each outcome according to the surface under the cumulative ranking curve was massage, acupressure, massage, and tuina for forced expiratory volume in 1 s (FEV1), forced vital capacity (FVC), FEV1/FVC, and 6 min walking distance, respectively. Conclusions: MTs such as massage, acupressure, and tuina have shown comparative benefits for lung function and exercise capacity in COPD. However, the methodological quality of the included studies was poor, and the head-to-head trial comparing the effects of different types of MTs for COPD patients was insufficient. Therefore, further high-quality RCTs are essential.

## 1. Introduction

Chronic obstructive pulmonary disease (COPD) is a common pulmonary disease characterized by persistent airflow limitation, which is usually associated with an enhanced chronic inflammatory response [1]. In addition, the harmful effects of toxic chemical particles or gases on the lungs often cause COPD; therefore, smoking is an important risk factor [1]. Epidemiological studies indicate that the prevalence of COPD is very common, ranging from 8% to 10% [2], and it causes significant economic and social burden worldwide [3].

The main therapeutic approaches for COPD include pharmacological treatment and lifestyle management such as cessation of smoking [4]. In pharmacological treatment for COPD, long-acting β2-agonist, long-acting muscarinic antagonists, inhaled corticosteroids, and bronchodilators, among others, may be used alone or in combination, depending on the patient’s condition or comorbidity [4].

COPD is a pathological condition associated with altered chest wall mechanics and musculoskeletal changes, and various manual therapies have been used as adjuvant therapy in combination with conventional medicine [5]. Although manual therapy is frequently used in clinical practice, some systematic reviews have pointed out that there is not enough evidence to support its therapeutic effect [6,7]. According to its theoretical basis, manual therapy can be classified as either based on Western medicine or on Eastern medicine. In the former case, it is applied based on the anatomical knowledge of the human body, such as manual therapy currently used in COPD, but in the latter case, manual therapy from a holistic perspective is applied within the body-mind-spirit model [8,9]. Previous studies that pointed out the lack of evidence for manual therapy for COPD have the limitation that they did not consider manual therapy based on Eastern medicine [6,7].

Various manual therapies take a common approach when the practitioner’s body comes in contact with the patient’s body; therefore, they will have a common expected effect along with the unique effect of each therapy, which leads to unique results for multiple outcomes of COPD. For example, advocators of Western manual therapy may explain that by improving musculoskeletal changes of altered chest wall mechanics, manual therapy can affect chest wall compliance of patients with COPD [5]. On the other hand, Eastern manual therapies are regarded as manual therapy combined with the traditional concept of meridian massage, and in this medical system, there is a view that both function and structure are systematically correlated [10]. Therefore, several Western and Eastern manual therapies can each have their own effectiveness for COPD, and comparative analysis of them can promote important individualized therapy in COPD management [11]. In addition, comparative analysis of Western and Eastern manual therapies for COPD could potentially help establish an integrative medical perspective for patients with COPD by combining the advantages of each.

Network meta-analysis (NMA) is a methodology that enables simultaneous comparison of various interventions at the same time [12]. By creating indirect evidence, it enables comparisons between interventions that are not directly compared by the existing clinical trials [12]. Therefore, this methodology can suggest the best intervention for each outcome to obtain optimal results for patients; further, it can be used to deduce clinical practice guidelines [13]. Until now, the effect of various manual therapies from the East and West on the outcome of COPD has not been comprehensively reviewed, and no attempts have been made to investigate the comparative effect of NMA. Therefore, this review aimed to compare the effectiveness and safety of several Western and Eastern manual therapies in COPD management. The results of this study will help to understand not only the clinical evidence of manual therapy for COPD more comprehensively but also understand the comparative effects of the presence or absence of a holistic approach (Western vs. Eastern).

## 2. Materials and Methods

The pre-registered protocol of this review can be found in OSF registries (doi:10.17605/OSF.IO/T2WM4). This systematic review complied with the Preferred Reporting Items for Systematic Reviews and Meta-Analyses (PRISMA) 2020 statement [14].

### 2.1. Data Sources and Search Strategy

To find relevant studies, a total of nine electronic databases, including Medline via PubMed, Excerpta Medica dataBASE (EMBASE) via Elsevier, the Cochrane Central Register of Controlled Trials, Allied and Complementary Medicine, Korean Studies Information Service System, Korea Citation Index, China National Knowledge Database, Wanfang Database, and Chinese Scientific Journals Database (CSJD-VIP), were comprehensively searched by one researcher (BL), without any limitations on language and publication status. The search date was 12 February 2021, and all published studies up to the search date were considered. In addition, we reviewed the reference list of included or related literature to find gray literature and requested advice from systematic review experts (Appendix A).

### 2.2. Eligibility Criteria

The inclusion criteria for this review were as follows: (1) Study type: Only randomized controlled trials (RCTs) were included in this review, while quasi-RCTs were excluded. (2) Types of participants: Adult patients (over 18 years of age) diagnosed with COPD were included in this study regardless of sex, COPD stage, and history of exacerbations. Patients with COPD having other significant diseases affecting the respiratory system, such as lung or other cancers, were excluded. Studies including people with COPD as well as other respiratory diseases (such as asthma or asthma COPD overlap syndrome) were also excluded. (3) Types of interventions: Western and Eastern manual therapies were included as interventions of interest, including manipulative therapy, joint mobilization, chiropractic, massage, reflexology, soft tissue therapy, muscle stretching, tuina, and acupressure passively applied using the practitioners’ hands. In this review, Western manual therapy was defined as manual therapy based on conventional Western anatomy. Specifically, manual therapy that mainly targets musculoskeletal changes of altered chest wall mechanics was considered Western manual therapy, which may include spinal manipulation, osteopathic manipulative treatment, manual diaphragm release technique, and soft tissue massage [5]. On the other hand, Eastern manual therapy was defined as manual therapy based on East Asian traditional medicine (EATM) theory such as meridian theory as well as conventional anatomy. Specifically, manual therapy targeting the meridian, a unique energy flow that connects the whole body in EATM, or based on a holistic perspective, was considered Eastern manual therapy, which may include tuina, reflexology, and acupressure [10]. Exercise therapy, self-treatment, active stretching, and therapies not performed by a practitioner were excluded. Additionally, acupressure with needles, seeds, or magnetic pieces on acupoints was also excluded. Although eligible treatments could be employed with or without other conventional interventions, it was imperative that the primary tested intervention applied manual therapy techniques. Oral or external herbal medicine, pharmacopuncture, acupuncture, moxibustion, qigong, taichi, and psychotherapy, which could not be considered conventional interventions, were excluded. (4) Types of controls: Comparators included no treatment, wait-list, sham treatment, routine pulmonary rehabilitation, medication, and other active controls. (5) Types of outcomes: The primary outcome was lung function parameters, such as forced expiratory volume in 1 s (FEV1), forced vital capacity (FVC), or FEV1/FVC, and exercise capacity, such as the 6 min walking distance (6MWD). Secondary outcomes were clinical symptoms such as the severity of dyspnea assessed using the Medical Research Council (MRC) dyspnea scale developed in England. Alternatively, other assessment tools such as patient-reported measures, self-assessment, and/or questionnaires could be used. In addition, quality of life measured using the COPD assessment test (CAT) was included as a secondary outcome. When CAT was not used, an alternate assessment tool, such as the St. George Respiratory Questionnaire (SGRQ), was allowed. Finally, the incidence of adverse events (AEs) or safety measurements was included as a secondary outcome. The outcome for the respiratory function was included in the analysis, but other outcomes such as constipation, anxiety, depression, and sleep disorder were not analyzed because they were not of interest to us. However, symptoms of sputum were considered, as they were indirectly related to respiratory function.

### 2.3. Study Selection

Two independent reviewers (CYK and BL) screened the titles and abstracts of the searched studies to determine their eligibility. Then, the full text of the screened studies was reviewed by two independent reviewers (CYK and BL) for inclusion. Discrepancies were resolved by discussion with a third researcher (KIK). EndNote X7 (Clarivate, Philadelphia, PA, USA), a reference management tool, was used in the study selection process.

### 2.4. Data Extraction

The data that were extracted from the eligible studies by two independent researchers (CYK and BL) were entered into a Microsoft Excel file. The following data were extracted: first author, country, information related to the risk of bias assessment, sample size, mean age and sex ratio of participants, the condition of COPD (acute exacerbations of COPD (AECOPD) and stable COPD), diagnostic criteria for COPD, pattern identification, details of intervention, methods of manual therapy, treatment duration, timing of assessment, outcomes, and results. Discrepancies were resolved by discussion with a third researcher (KIK).

### 2.5. Risk of Bias Assessment

The risk of bias of the included studies was assessed according to the Cochrane Handbook version 5.1.0. assessment tool by two independent researchers (CYK and BL) [15]. In the Cochrane’s risk of bias tool, domains for random sequence generation, allocation concealment, blinding of participants and personnel, blinding of outcome assessment, incomplete outcome data, selective reporting, and other sources of bias are evaluated as “low,” “high,” or “unclear” [15]. The risk of bias summary figure was produced using the RevMan Software version 5.4 (The Cochrane Collaboration, London, England). Discrepancies were resolved by discussion with a third researcher (KIK).

### 2.6. Data Analysis and Synthesis

The baseline characteristics and outcomes of all the included studies were analyzed descriptively.

#### 2.6.1. Conventional Pair-Wise Meta-Analysis

When there was adequate homogeneous data, quantitative synthesis was performed using RevMan 5.4 (The Cochrane Collaboration, London, England). Dichotomous data were presented as risk ratio (RR) with 95% confidence interval (CI), and continuous data were reported as mean difference (MD) with 95% CI. Heterogeneity between the studies in terms of effect measures was assessed using both the χ² test and the I² statistic. I² values of ≥50% and ≥75% were considered indicative of substantial and considerable heterogeneity, respectively. Due to the nature of non-pharmacological therapies, which are the interventions of interest in this review, it was difficult to guarantee the homogeneity of the implementation of the interventions, so we applied the random-effects model to meta-analyses [16]. When a sufficient number of studies (≥10) were included in each meta-analysis, publication bias was evaluated using a funnel plot.

#### 2.6.2. Network Meta-Analysis

NMA was performed on primary outcomes to provide both direct and indirect evidence. Routine care (ROC) was used as the reference treatment. NMA based on the frequent framework was carried out using mvmeta and network packages in Stata software version 16 (StataCorp, College Station, TX, USA). Inconsistency was assessed using the node-splitting method and the design-by-treatment interaction model, and a random-effects NMA model was selected. Potential publication bias was assessed using a net funnel plot, provided a sufficient number of studies (≥10) were included. In addition, we examined the surface under the cumulative ranking curve (SUCRA) statistic to identify the best treatment. The overall NMA method in this review followed that of Shim et al. (2017) [17].

### 2.7. Dealing with Missing Data

The authors contacted the corresponding author via email regarding any unclear information in the concerned study. If the data were still insufficient after contacting the corresponding author or if contact was not possible, it was analyzed using the available data.

## 3. Results

### 3.1. Study Selection

Through database searching, a total of 2623 articles were searched, and no studies were identified through other sources. After removing 843 duplicates, the titles and abstracts of 1780 articles were screened for first inclusion, and 1714 studies were excluded. After assessing full-text of the remaining 66 articles, 36 studies including eight only abstract available with no details, one case series, four quasi-RCTs, two review articles, 10 not meeting intervention criteria, five without outcomes of interest, four using duplicated data, and two unavailable full-text were excluded (Appendix A). Finally, total 30 RCTs [18,19,20,21,22,23,24,25,26,27,28,29,30,31,32,33,34,35,36,37,38,39,40,41,42,43,44,45,46,47] were included in qualitative synthesis, and 21 RCTs [19,20,22,23,24,27,28,29,30,31,33,35,36,38,41,42,43,44,45,46,47] were included in meta-analysis (Figure 1).

### 3.2. Study Characteristics

In total, 19 studies [19,22,23,24,25,26,27,28,29,30,33,35,38,41,43,44,45,46,47] were published in China, two were published in Italy [18,42], Poland [31,34], and the United Kingdom [20,39], and one was published in the USA [36], Brazil [37], Australia [21], Turkey [32], and Taiwan [40]. There were 11 studies [18,20,21,31,32,34,36,37,38,42,47] using Western manual therapy, of which four [18,34,36,42] were related to manipulation, four [21,31,32,47] were regarding massage, and one study each was related to release technique [37], manual chest technique [20], and manual percussion [38]. Nineteen studies [19,22,23,24,25,26,27,28,29,30,33,35,39,40,41,43,44,45,46] used Eastern manual therapy, of which 10 [22,23,24,28,30,33,40,41,45,46] were related to acupressure, five [25,26,27,29,39] pertained to foot reflexology, and four [19,35,43,44] were regarding tuina. As for the COPD stage, 16 studies [18,19,25,26,27,28,29,30,34,35,36,37,39,42,43,45] targeted stable COPD participants, eight studies [20,22,23,24,31,33,38,46] were on AECOPD, and six studies [21,32,40,41,44,47] did not specify the stage. Nine studies [22,23,24,28,29,30,35,41,45] targeted participants with specific pattern identification, of which four focused on participants with phlegm-heat [22], phlegm turbidity [23,41], or phlegm-blood stasis [24] obstructing the lung, and five [28,29,30,35,45] targeted dual deficiency of lung–spleen or lung–kidney. Twelve studies [18,20,21,22,23,24,32,34,36,37,42,45] were approved by the Institutional Review Board prior to study commencement, and 19 studies [18,20,21,22,23,24,25,26,29,32,34,35,36,37,40,41,42,45,47] received consent forms from the participants (Table 1 and Table 2).

### 3.3. Risk of Bias Assessment

Thirteen studies [22,23,24,25,26,27,28,29,34,37,40,42,43] that used an appropriate random sequence generation method such as random number tables were evaluated as having a low risk of selection bias, and three studies [21,32,37] that properly concealed allocation using an opaque sealed envelope were also evaluated as having a low risk of selection bias. Three studies [34,36,37] that reported that the practitioners who were not blinded were at high risk of performance bias, and one study [42] that reported that both participants and personnel were blinded was evaluated as having a low risk of performance bias. Five studies [20,21,36,37,42] reporting blindness of outcome assessors were evaluated as having a low risk of detection bias. Three studies [32,39,43] that performed per-protocol analysis without specifying the reason for dropout were evaluated as having a high risk of attrition bias. Three studies [18,39,43] did not report raw data, and four studies [23,24,41,46] that did not report pulmonary function-related outcomes were evaluated as having a high risk of reporting bias. One study [28] without baseline characteristic data and one study [34] with cross-over design was evaluated as having a high risk of other potential biases (Figure 2).

### 3.4. Effectiveness and Safety of Manual Therapies Using Pair-Wise Meta-Analysis

Manipulation showed no differences compared with sham in lung functions (FEV1: MD 0.23 L, 95% CI from −0.12 to 0.58; FVC: MD −0.02 L, 95% CI from −0.57 to 0.53; FEV1/FVC: MD 3.01%, 95% CI from −6.90 to 12.92), exercise capacity (6MWD: MD 64.80 m, 95% CI from −12.94 to 142.54), and incidence of AE (RR 0.50, 95% CI: from 0.11 to 2.38). Additional massage significantly improved FEV1/FVC (MD 20.00%, 95% CI from 15.46 to 24.54) and total effective rate (TER) calculated using the severity of respiratory symptoms (RR 1.17, 95% CI from 1.00 to 1.38), compared with ROC alone. However, there were no differences between them in the FEV1 (MD 0.68 L, 95% CI from −0.62 to 1.99), 6MWD (MD 56.20 m, 95% CI from −8.18 to 120.58), and incidence of AE (RR 8.89, 95% CI from 0.48 to 165.55). When comparing additional acupressure with ROC alone, although there was no difference in FEV1 (MD 0.05 L, 95% CI from −0.24 to 0.34) and FEV1/FVC (MD 0.84%, 95% CI from −4.60 to 2.27), other outcomes including FVC (MD 0.33 L, 95% CI from 0.17 to 0.49), 6MWD (MD 14.38 m, 95% CI from 3.71 to 25.05), TER based on the respiratory symptom (RR 1.14, 95% CI from 1.06 to 1.23), sputum secretion (MD −5.31 mL, 95% CI from −6.00 to −4.62), SpO2 (MD 3.44%, 95% CI from 1.64 to 5.23), PaO2 (MD 13.38 mmHg, 95% CI from 9.16 to 17.60), PaCO2 (MD −8.91 mmHg, 95% CI from −12.09 to −5.72), and SaO2 (MD 9.10%, 95% CI from 5.29 to 12.91) significantly improved. Additional tuina significantly improved FEV1/FVC (MD 2.65%, 95% CI from 0.10 to 5.20), and 6MWD (MD 49.53 m, 95% CI from 27.05 to 72.00) compared with ROC alone, although there were no differences in FEV1 (MD 0.10 L, 95% CI from −0.05 to 0.25), FVC (MD 0.26 L, 95% CI from −0.05 to 0.58), and TER based on the respiratory symptom (RR 1.10, 95% CI from 0.94 to 1.28). When conducting foot reflexology in addition to ROC, 6MWD (MD 36.08 m, 95% CI from 8.45 to 63.71), and TER based on the respiratory symptom (RR 2.13, 95% CI from 1.09 to 4.16) significantly improved compared with ROC alone (Appendix A).

### 3.5. Comparative Effectiveness of Manual Therapies Using NMA

NMA was possible only for the outcomes of FEV1, FVC, FEV1/FVC, and 6MWD. Therefore, pair-wise meta-analysis was performed for other outcomes because the network had no degrees of freedom for heterogeneity due to the small number of studies included. Figure 3 shows the network map of the interventions belonging to each NMA.

#### 3.5.1. Lung Function

In FEV1, only additional massage showed significantly better results compared to ROC alone (MD 0.74 L, 95% CI 0.08 to 1.40). In FVC, additional acupressure resulted in significant improvement while tuina showed borderline better results compared to ROC alone (MD 0.33 L, 95% CI from 0.17 to 0.47; MD 0.26 L, 95% CI from −0.05 to 0.58) (Table 3). In FEV1/FVC, additional massage showed significantly better results not only compared to ROC alone (MD 20.00%, 95% CI from 12.16 to 27.84) but also compared to additional acupressure (MD 19.18%, 95% CI from 10.23 to 28.13) and additional tuina (MD 16.99%, 95% CI from 7.86 to 26.13). No statistically significant differences between the interventions were observed (Table 4). The most optimal treatment based on SUCRA in FEV1 and FEV1/FVC was additional massage, followed by additional tuina and acupuncture, and ROC. Furthermore, the most optimal treatment in FVC was additional acupressure, followed by additional tuina and ROC (Table 5).

#### 3.5.2. Exercise Capacity

In 6MWD, additional tuina showed significantly better results compared to ROC alone (MD 49.49 m, 95% CI from 25.60 to 73.38) and borderline better results compared to additional acupressure (MD 35.11 m, 95% CI from −1.03 to 71.26). Furthermore, additional foot reflexology showed borderline significant results compared to ROC alone (MD 36.08 m, 95% CI from −1.14 to 73.30) (Table 4). The most optimal treatment based on SUCRA in 6MWD was additional tuina, followed by additional massage, foot reflexology, acupressure, and ROC (Table 5).

## 4. Discussion

### 4.1. Summary of Evidence

This systematic review attempted to estimate the comparative effectiveness of Western and Eastern manual therapies for COPD patients based on a total of 30 RCTs [18,19,20,21,22,23,24,25,26,27,28,29,30,31,32,33,34,35,36,37,38,39,40,41,42,43,44,45,46,47]. Data for five interventions, including manipulation, massage, acupressure, tuina, and foot reflexology, were obtained from the pair-wise meta-analysis results. Additional massage (FEV/FVC), acupressure (FVC, 6MWD), tuina (FEV1/FVC, 6MWD), and foot reflexology (6MWD) showed significantly improved results compared to ROC alone in one or more outcomes of lung functions and/or exercise capacity. However, manipulation did not show significantly better results (FEV1, FVC, FEV1/FVC, 6MWD) compared to sham treatment. In addition, there was evidence that additional acupressure and tuina could significantly improve the quality of life of COPD patients (CAT and SGRQ), although meta-analysis could not be carried out because there was only one study that evaluated this outcome. Additional acupressure could significantly improve some objective outcomes of COPD patients, including sputum secretion, SpO2, PaO2, PaCO2, and SaO2. There were no interventions that significantly differed in the incidence of AEs compared to the controls. The number of interventions included in the NMA for FEV1, FVC, FEV1/FVC, and 6MWD was four, three, four and five, respectively. According to the results, only additional massage for FEV1 and only additional acupressure for FVC showed significantly better results than ROC. On the other hand, additional massage for FEV1/FVC showed significantly better results than ROC, acupressure, and tuina. Only additional tuina showed significantly better results for 6MWD than ROC. However, the comparative effect of foot reflexology was not significant for any outcome. The optimal treatment for each outcome according to SUCRA was massage, acupressure, massage, and tuina for FEV1, FVC, FEV1/FVC, and 6MWD, respectively.

The methodological quality of the included studies was generally poor. Limitations of methodological quality were found throughout the evaluated domains in the Cochrane’s risk of bias tool, and more than half of the studies were evaluated as having an unclear or high risk of bias in relation to random sequence generation, allocation concealment, and blinding procedures. This suggests that the study results derived from the included studies may have been influenced by the placebo effect or overestimated.

### 4.2. Clinical Implications

Although the main therapeutic approaches for COPD are pharmacological approaches and lifestyle management [4], manual therapy is considered a promising adjuvant therapy [5]. In this review, various types of manual therapies were categorized as therapies derived from the East or West according to their origins, and the most effective manual therapy for individual outcomes related to COPD was explored through the NMA methodology. Although with limited certainty, some clinical evidence indicated that massage was the most effective treatment for FEV1 and FEV1/FVC, acupressure for FVC, and tuina for 6MWD. Since manual therapies are generally used as a complement to conventional treatment for COPD in clinical settings, the findings of this review suggest that it may be helpful to select a specific manual therapy method according to the individual patient’s characteristics and target symptoms.

COPD is a long-standing problem, and the development of non-pharmacological therapies to improve the quality of life of COPD patients is important [48]. In this review, manual therapies that showed significant improvement in some outcomes of the quality of life in COPD patients were acupressure and tuina belonging to Eastern manual therapy. These therapies may not only affect lung function or exercise capacity in COPD patients but may also help improve other disturbing symptoms, including pain [49], insomnia [50,51], and fatigue [50], as seen in previously published studies, thereby contributing to the improvement in the quality of life of COPD patients.

### 4.3. Limitations

This systematic review attempted to conduct a comprehensive review of the various types of manual therapies utilized for COPD and to investigate its comparative effectiveness on lung function and exercise capacity of COPD patients using the NMA methodology. However, the results of this review should be interpreted considering the following limitations.

First, given the heterogeneity of interventions investigated, the number of studies that were included in this review (30 in total) was not sufficient to provide strong evidence through quantitative synthesis. In addition, given that most of the included studies had small sample sizes, there is a possibility that the findings of this review were greatly influenced by small-study effects [52]. Second, the quality of the included studies was poor overall. In particular, as aforementioned, the results from these studies may have been influenced by placebo effects or could have been overestimated, as random sequence generation, allocation concealment, and blinding procedures were described unclearly or with a high risk of bias. Third, in our prior protocol, evaluation of publication bias of the included studies was planned using funnel plots and net funnel plots. However, the lack of included studies consequently made it impossible to visually evaluate publication bias using funnel plots. This implies that we cannot rule out the possibility that the results reported in the studies included in this review may be biased. Finally, comparisons between manual therapies performed in this review primarily came from NMA, and the data are lacking in conventional pair-wise meta-analysis. That is, the head-to-head trial comparing the comparative effects of different types of manual therapies for COPD patients in conventional RCTs was insufficient. In particular, head-to-head trials between Western manual therapy and Eastern manual therapy, one of the rationales of this systematic review, did not exist. Although the NMA methodology enables indirect comparison between interventions that have not previously been directly compared with each other [12], the overall poor methodological quality of the included studies suggests that large-scale, high-quality head-to-head trials can provide more reliable results. Given that various manual therapies are being used and studied for COPD patients in both East and West, robust clinical trials evaluating the comparative effectiveness of these treatments may be of interest to future researchers.

## 5. Conclusions

This systematic review estimated the comparative effectiveness of Western and Eastern manual therapies for patients with COPD using the NMA methodology. The optimal treatment for each outcome according to SUCRA was massage, acupressure, massage, and tuina for FEV1, FVC, FEV1/FVC, and 6MWD, respectively. However, the methodological quality of the included studies was generally poor, and the head-to-head trial comparing different types of manual therapies for COPD patients was inadequate. Given the complementary role and promise of manual therapies in the treatment of patients with COPD, high-quality RCTs in this area should be implemented in the future.

## Figures and Tables

**Figure 1 healthcare-09-01127-f001:**
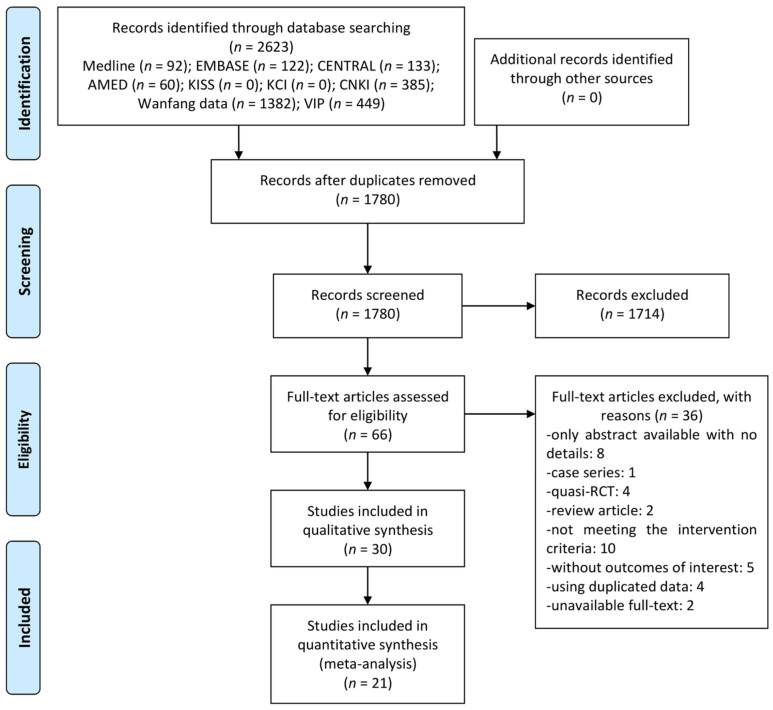
A PRISMA flow diagram of the literature screening and selection process. AMED—Allied and Complementary Medicine Database; CENTRAL—Cochrane Central Register of Controlled Trials; CNKI—China National Knowledge Infrastructure; KCI—Korea Citation Index; KISS—Korean Studies Information Service System; RCT—randomized controlled trial.

**Figure 2 healthcare-09-01127-f002:**
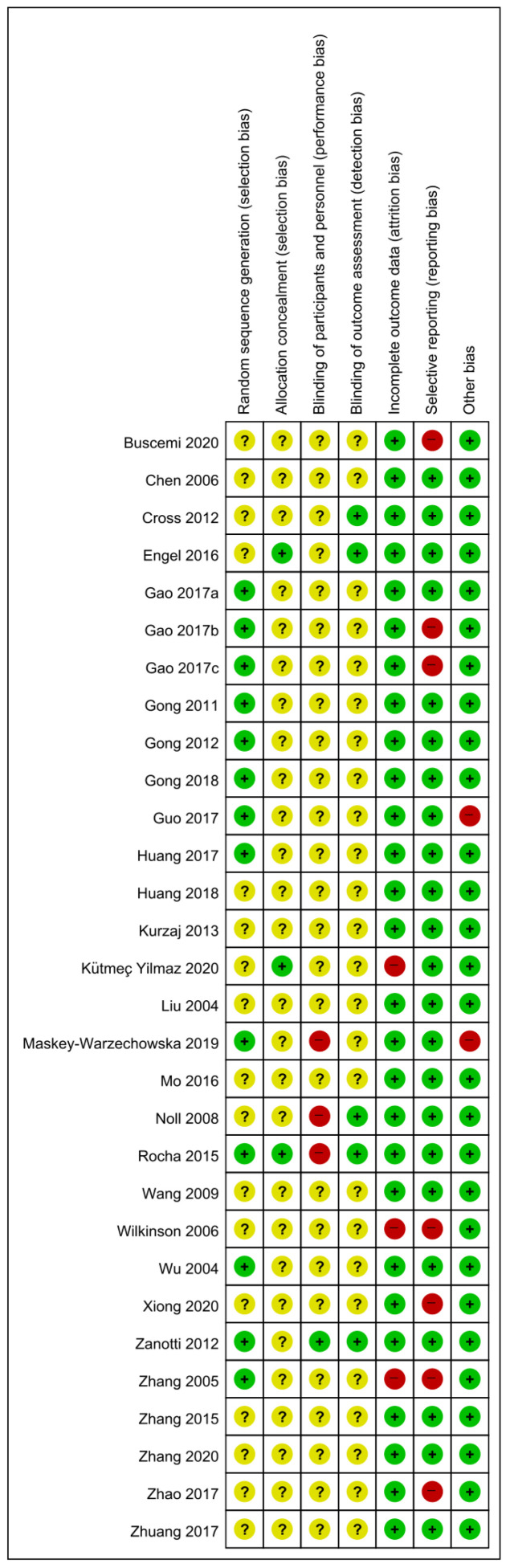
Risk of bias for all included studies. Low, unclear, and high risk, respectively, are represented with the following symbols: “+”, “?”, and “−”.

**Figure 3 healthcare-09-01127-f003:**
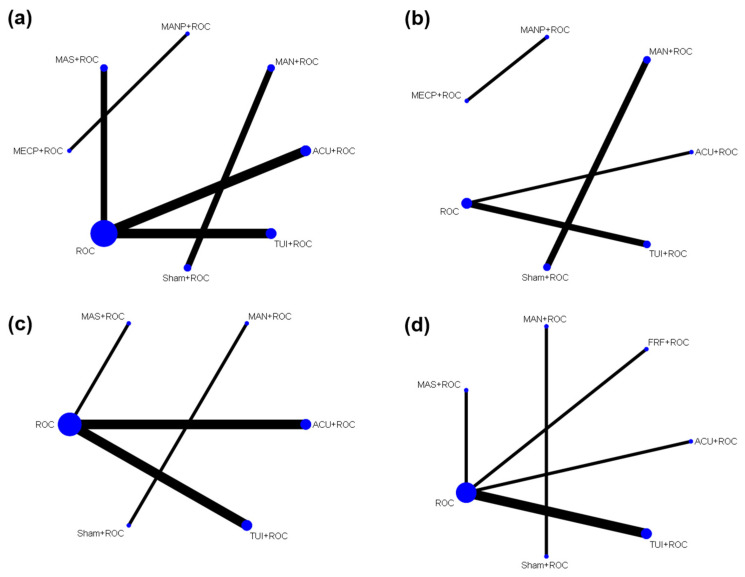
Network map of (**a**) FEV1 (L), (**b**) FVC (L), (**c**) FEV1/FVC (%), and (**d**) 6 min walking distance. ACU—acupuncture; FRF—foot reflexology; MAN—manipulation; MANP—manual percussion; MAS—massage; MECP—mechanical percussion; ROC—routine care; TUI—tuina.

**Table 1 healthcare-09-01127-t001:** Characteristics of included studies using Western manual therapy.

First Author (Year), Country	Sample Size (Included →Analyzed)	Mean Age (Year)	Diagnosis	(A) Treatment Intervention	(B) Control Intervention	Duration of Treatment/Follow-Up	Outcome of Interest
Stable COPD							
Noll (2008), USA	35(18:17)→35(18:17)	(A) 69.6 ± 6.6 (B) 72.2 ± 7.1	Known COPD history, FEV1/FVC < 70%	Osteopathic manipulative treatment	Sham (light touch)	20 min one session/none	1. FEV1 (L); 2. FVC (L); 3. FEV1/FVC (%); 4. Adverse events
Zanotti (2012), Italy	20(10:10)→20(10:10)	(A) 63.5 ± 4.7 (B) 64.2 ± 5.5	GOLD	Osteopathic manipulative treatment	Sham (light touch)	45 min once a week for 4 weeks/none	1. VC (L); 2. FEV1 (L); 3. FVC (L); 4. 6MWD (m); 5. Adverse events
Maskey-Warzechowska (2019), Poland	38(19:19)→38(19:19)	68	Severe-very severe (FEV1 < 50%), GOLD 2016	Osteopathic manipulative treatment	Sham	25 min one session/none	Note. Only median (IQR) value was reported. 1. FEV1 (L, %pv); 2. FVC (L, %pv); 3. FEV1/FVC (%); 4. VAS (dyspnea); 5. Adverse events
Buscemi (2020), Italy	32→32	71	Moderate-severe COPD	Osteopathic manipulative treatment + (B)	Conventional pharmacotherapy	once a week for 8 weeks/15 days	Note. Raw data were not reported except adverse events. 1. FVC (L); 2. FEV1 (L); 3. CAT; 4. 6MWD (m); 5. Adverse events
Rocha (2015), Brazil	20(11:9)→19(10:9)	(A) 71 ± 5 (B) 71 ± 6	GOLD 2011	Manual diaphragm release technique	Sham (light touch)	6 times on non-consecutive days within 2 weeks/none	Note. Only change value was reported. 1. 6MWD (m)
AECOPD							
Kurzaj (2013), Poland	30(20:10)→30(20:10)	(A) 57 (B) 55	NR	Massage + (B)	Basic physiotherapy (Respirometric training)	30 min daily for 6 days/none	1. FEV1 (L, %pv); 2. 6MWD (m); 3. MRC; 4. BODE index
Cross (2012), UK	522(258:264)→372(186:186)	(A) 69.08 ± 9.85 (B) 69.58 ± 9.51	NR	Manual chest technique + (B)	Breathing technique	1–41 min, total 1–21 sessions/6 mon	Note. Raw data of sputum volume and SpO2 were not reported. 1. SGRQ; 2. Breathlessness Cough and Sputum Scale; 3. EQ-5D; 4. EQ-VAS; 5. Sputum volume (mL); 6. SpO2; 7. Hospitalization period; 8. Adverse events
Wang (2009), China	120(60:60)→120(60:60)	NR	NR	Manual percussion	Mechanical percussion	twice a day for 7 days/none	1. Sputum excretion (mL); 2. SpO2; 3. Hospitalization period; 4. Time to improvement/disappearance of cough, dyspnea, and sputum sound in lungs; 5. FVC (L); 6. FEV1 (L)
Unclear COPD							
Engel (2016), Australia	33(9:9:15)→31(8:8:15)	(A1) 67.6 ± 3.5 (A2) 65.0 ± 4.1 (B) 64.5 ± 4.1	NR	(A1) Massage + (B) (A2) Massage + Spinal manipulation + (B)	Pulmonary rehabilitation	twice a week for 8 weeks/none	Note. Only change value was reported. 1. FEV1 (L); 2. FVC (L); 3. SGRQ; 4. 6MWD (m); 5. Adverse events
Zhuang (2017), China	70(35:35)→70(35:35)	(A) 64.98 ± 4.98 (B) 65.23 ± 5.25	NR	Massage + (B)	Routine care (medication, exercise education, diet education, etc.)	NR/none	1. FEV1 (L); 2. FEV1/FVC (%); 3. TER (respiratory symptom)
Kütmeç Yilmaz (2020), Turkey	91(49:42)→58(28:30)	70.6	NR	Back massage + (B)	Routine care	15 min, daily for 4 days/none	Note. Only median (IQR) value was reported. 1. SpO2

Abbreviations: AECOPD—acute exacerbations of chronic obstructive pulmonary disease; BODE—body-mass index, obstruction of airways, dyspnea, exercise capacity; CAT—chronic obstructive pulmonary disease assessment test; COPD—chronic obstructive pulmonary disease; EQ-VAS—EuroQol-visual analogue scale; EQ-5D—EuroQol-5 dimension; FEV1—forced expiratory volume in one second; FVC—forced vital capacity; GOLD—global initiative for chronic obstructive lung disease; IQR—interquartile range; MRC—medical research council dyspnea scale; NR—not recorded; pv—predicted value; SGRQ—St. George respiratory questionnaire; TER—total effective rate; VAS—visual analogue scale; VC—vital capacity; 6MWD—6 min walking distance.

**Table 2 healthcare-09-01127-t002:** Characteristics of included studies using Eastern manual therapy.

First Author (Year), Country	Sample Size (Included→Analyzed)	Mean Age (Year)	Diagnosis	(A) Treatment Intervention	(B) Control Intervention	Duration of Treatment/Follow-Up	Outcome of Interest
Stable COPD							
Guo (2017), China	200(100:100)→200(100:100)	NR	Criteria used by associations or guidelines in China	Acupressure + (B)	Basic physiotherapy (Respirometric training)	2–3 min per acupoint, once a day for 6 mon/none	1. FEV1 (L); 2. FEV1/FVC (%); 3. SGRQ
Huang (2018), China	68(34:34)→68(34:34)	(A) 52.43 ± 3.96 (B) 54.43 ± 1.27	COPD	Acupressure + (B)	Basic physiotherapy (Respirometric training)	2 min per acupoint twice a day for 3 mon/none	1. FEV1 (L, %pv); 2. FEV1/FVC (%); 3. 6MWD (m); 4. CAT
Zhang (2020), China	90(45:45)→90(45:45)	(A) 67.46 ± 5.23 (B) 67.85 ± 5.62	Criteria used by associations or guidelines in China	Acupressure + (B)	Basic physiotherapy (Respirometric training)	2–3 min per acupoint once a day for 6 mon/none	1. FVC (L); 2. FEV1 (L); 3. FEV1/FVC (%)
Wilkinson (2006), UK	14(7:7)→14(7:7)	(A) 77 (B) 75	Moderate–severe COPD	Foot reflexology	No intervention	50 min once a week for 4 weeks/none	Note. Raw data were not reported. 1. Quality of life (questionnaire); 2. SpO2
Gong (2011), China	60(30:30)→60(30:30)	(A) 67.03 ± 9.48 (B) 69.93 ± 8.18	Criteria used by associations or guidelines in China	Foot reflexology + (B)	Health education	30 min once a day for 3 mon/none	Note. Only change value was reported. 1. SGRQ; 2. FEV1 (L, %pv); 3. FEV1/FVC (%)
Gong (2012), China	60(30:30)→60(30:30)	(A) 67.03 ± 9.48 (B) 69.93 ± 8.18	Criteria used by associations or guidelines in China	Foot reflexology + (B)	Health education	30 min once a day for 3 mon/none	Note. On 6MWD and MRC, only change values were reported. 1. 6MWD (m); 2. MRC; 3. TER (respiratory symptom)
Huang (2017), China	60(30:30)→59(29:30)	(A) 69.52 ± 4.31 (B) 69.37 ± 4.56	Criteria used by associations or guidelines in China	Foot reflexology + (B)	Health education	30 min once a day for 6 mon/none	1. 6MWD (m)
Gong (2018), China	60(30:30)→59(29:30)	(A) 69.52 ± 4.31 (B) 69.37 ± 4.56	Criteria used by associations or guidelines in China	Foot reflexology + (B)	Health education	30 min once a day for 6 mon/none	Note. Only change value was reported. 1. BODE index; 2. 6MWD (m); 3. modified MRC; 4. FEV1 (%)
Zhang (2005), China	66(33:33)→63(31:32)	(A) 68.3 ± 6.79 (B) 67.7 ± 7.92	Criteria used by associations or guidelines in China	Tuina	Basic physiotherapy (Respirometric training)	20 min 3 times a week for 3 mon/none	1. FVC (%); 2. FEV1 (L); 3. FEV1/FVC (%); 4. 6MWD (m); 5. TER (SGRQ); 6. Quality of life (questionnaire)
Chen (2006), China	30(15:15)→30(15:15)	(A) 69.12 ± 6.21 (B) 67.63 ± 7.01	Criteria used by associations or guidelines in China	Tuina + (B)	Conventional pharmacotherapy	20 min 5 times a week for 8 weeks/none	1. TER (dyspnea); 2. FEV1/FVC (%); 3. FEV1 (L); 4. FVC (L); 5. 6MWD (m)
Mo (2016), China	60(30:30)→57(29:28)	(A) 56.5 ± 6.2 (B) 58.4 ± 5.6	Criteria used by associations or guidelines in China	Tuina + (B)	Conventional pharmacotherapy	6 times a week for 4 weeks/none	1. CAT; 2. 6MWD (m); 3. Adverse events
AECOPD							
Liu (2004), China	127(64:63)	(A) 65.33 ± 4.44 (B) 64.49 ± 5.63	Criteria used by associations or guidelines in China	Acupressure + (B)	Routine care	1hr once a day for 7 days/none	1. TER (respiratory symptom); 2. Time to improve cough, sputum, and dyspnea; 3. PaO2; 4. PaCO2
Gao (2017a), China	60(30:30)→60(30:30)	(A) 66.65 ± 3.70 (B) 68.77 ± 4.28	Criteria used by associations or guidelines in China	Acupressure + (B)	Routine care	1 min per acupoint for 10 min twice a day for 7 days/none	1. TER (respiratory symptom); 2. Symptom score (cough, sputum, wheezing, shortness of breath); 3. PaO2; 4. PaCO2; 5. SaO2
Gao (2017b), China	60(30:30)→60(30:30)	(A) 70.5 ± 4. (B) 68.5 ± 4.7	Criteria used by associations or guidelines in China	Acupressure + (B)	Routine care	20 min twice a day for 7 days/none	1. Sputum excretion (mL); 2. SpO2; 3. PaO2; 4. PaCO2
Gao (2017c), China	60(30:30)→60(30:30)	(A) 69.36 ± 5.65 (B) 70.84 ± 4.76	Criteria used by associations or guidelines in China	Acupressure + (B)	Routine care	15 min twice a day for 7 days/none	1. Symptom score (cough, sputum, asthma, dyspnea); 2. PaO2; 3. PaCO2; 4. SaO2; 5. TER (respiratory symptom); 6. Hospitalization period
Zhao (2017), China	58(29:29)→58(29:29)	(A) 67.5 ± 3.6 (B) 68.5 ± 4.1	NR	Acupressure + (B)	Routine care	10 min twice a day for 7 days/none	1. Sputum excretion (mL); 2. SpO2; 3. PaO2; 4. PaCO2
Unclear COPD							
Wu (2004), Taiwan	44(22:22)→44(22:22)	73 ± 9.7	NR	Acupressure	Sham (unrelated acupoint)	16 min five times a week for 4 weeks/none	Note. Only change value was reported. 1. 6MWD (m); 2. SpO2
Xiong (2020), China	120(60:60)→120(60:60)	(A) 50.89 ± 4.58 (B) 55.72 ± 4.54	NR	Acupressure + (B)	Routine care	30 min twice a day for 1 mon/none	1. TER (respiratory symptom); 2. Sputum excretion (mL)
Zhang (2015), China	80(40:40)→80(40:40)	(A) 45 ± 2.5 (B) 43 ± 3.4	Criteria used by associations or guidelines in China	Tuina + (B)	Routine care (medication)	~25 min five times a week for 7 weeks/none	1. TER (respiratory symptom); 2. FEV1/FVC (%); 3. FEV1 (L); 4. FVC (L)

Abbreviations: AECOPD—acute exacerbations of chronic obstructive pulmonary disease; BODE—body-mass index, obstruction of airways, dyspnea, exercise capacity; CAT—chronic obstructive pulmonary disease assessment test; COPD—chronic obstructive pulmonary disease; FEV1—forced expiratory volume in one second; FVC—forced vital capacity; MRC—medical research council dyspnea scale; NR—not recorded; pv—predicted value; SGRQ—St. George respiratory questionnaire; TER—total effective rate; 6MWD—6 min walking distance.

**Table 3 healthcare-09-01127-t003:** Network league table of FEV1 (L) (right upper part) and FVC (L) (left lower part).

ROC	0.04 (−0.48, 0.55)	0.74 (0.08, 1.40)	0.09 (−0.46, 0.65)
0.33 (0.17, 0.49)	ACU + ROC	0.70 (−0.13, 1.54)	0.05 (−0.70, 0.81)
		MAS + ROC	−0.65 (−1.51, 0.21)
0.26 (−0.05, 0.58)	−0.07 (−0.42, 0.28)		TUI + ROC

Note: Data are presented in mean difference (95% confidence interval). The result underlined meant it had statistical significance. Abbreviations. ACU—acupuncture; FEV1—forced expiratory volume in one second; FVC—forced vital capacity; MAS—massage; ROC—routine care; TUI—tuina.

**Table 4 healthcare-09-01127-t004:** Network league table of FEV1/FVC (%) (right upper part) and 6 min walking distance (m) (left lower part).

ROC	0.82 (−3.50, 5.14)		20.00 (12.16, 27.84)	3.01 (−1.68, 7.69)
14.38 (−12.74, 41.50)	ACU + ROC		19.18 (10.23, 28.13)	2.19 (−4.18, 8.56)
36.08 (−1.14, 73.30)	21.70 (−24.35, 67.75)	FRF + ROC		
56.20 (−12.84, 125.24)	41.82 (−32.36, 116.00)	20.12 (−58.31, 98.55)	MAS + ROC	−16.99 (−26.13, −7.86)
49.49 (25.60, 73.38)	35.11 (−1.03, 71.26)	13.41 (−30.81, 57.64)	−6.71 (−79.77, 66.35)	TUI + ROC

Note: Data are presented in mean difference (95% confidence interval). The result underlined meant it had statistical significance. Abbreviations. ACU—acupuncture; FEV1—forced expiratory volume in one second; FRF—foot reflexology; FVC—forced vital capacity; MAS—massage; ROC—routine care; TUI—tuina.

**Table 5 healthcare-09-01127-t005:** SUCRA for interventions on each outcome.

Interventions	FEV1 (L)	FVC (L)	FEV1/FVC (%)	6MWD (m)
ROC	27.9	2.5	15.4	6
ACU + ROC	35.1	82.4	29.9	30
FRF + ROC				59.5
MAS + ROC	95.6		100	76.5
TUI + ROC	41.4	65.1	54.7	78

Abbreviations. ACU—acupuncture; FEV1—forced expiratory volume in one second; FRF—foot reflexology; FVC—forced vital capacity; MAS—massage; ROC—routine care; TUI—tuina; 6MWD—6 min walking distance.

## Data Availability

The data presented in this study are available in the article and Appendix A.

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
