# Peer review of "Comparative Effectiveness of Western and Eastern Manual Therapies for Chronic Obstructive Pulmonary Disease: A Systematic Review and Network Meta-Analysis"

_healthcare, 2021, doi:10.3390/healthcare9091127_

Round 1

Reviewer 1 Report

COPD is a very complex disease and currently there are limited pharmacological and surgical interventions that could relief patient conditions. In this context, other interventions aimed at improving the COPD patient respiratory capacity and their quality of life are absolutely relevant.

Here the manuscript from Kwon et al. recapitulate the pros and cons of the manual therapies comparing different approaches coming from Eastern and Western countries.

The review is very well structured and provide useful insight and perspective to further explore this field.

Author Response

Comment

COPD is a very complex disease and currently there are limited pharmacological and surgical interventions that could relief patient conditions. In this context, other interventions aimed at improving the COPD patient respiratory capacity and their quality of life are absolutely relevant.

Here the manuscript from Kwon et al. recapitulate the pros and cons of the manual therapies comparing different approaches coming from Eastern and Western countries.

The review is very well structured and provide useful insight and perspective to further explore this field.

Author response

  • Thank you for the careful review of our manuscript.

Reviewer 2 Report

The manuscript entitled “Comparative effectiveness of Western and Eastern manual therapies for chronic obstructive pulmonary disease: A systematic review and network meta-analysis” aims to compare the effectiveness and safety of several Western and Eastern manual therapies in COPD management. I have revised the manuscript and provide some comments and suggestions which I hope the Authors will revise. The following comments are offered to help strengthen the manuscript:

Abstract section:

In the abstract it is not clearly justified the need of comparison of Western and Eastern manual therapies.

Introduction section:

In the introduction it is not clearly justified the need of comparison of Western and Eastern manual therapies.

Methods section:

Defined inclusion criteria include: “Western and Eastern manual therapies”. A more exhaustive definition of the manual therapies included in each group should be provided, indicating the names of the manual therapies included in each group and the reasons.

In Supplement 3. Results of pairwise meta-analysis.

  1. FEV1 (L): should be revised as data are not clearly view. Maybe range -20 to 20 should be modified.
  2. FEV1 (%pv): what this metanalysis add? There is not an overall result only the information provided study by study.

The same occur in several of the next analysis. At least two studies should be included in each analysis. This does not occur for “4. FVC (%pv)”

In the metanalysis Western and Eastern manual therapies are not compared.

Author Response

Comment 1

The manuscript entitled “Comparative effectiveness of Western and Eastern manual therapies for chronic obstructive pulmonary disease: A systematic review and network meta-analysis” aims to compare the effectiveness and safety of several Western and Eastern manual therapies in COPD management. I have revised the manuscript and provide some comments and suggestions which I hope the Authors will revise. The following comments are offered to help strengthen the manuscript:

Author response

  • Thank you for the careful review of our manuscript.

Comment 2

Abstract section:

In the abstract it is not clearly justified the need of comparison of Western and Eastern manual therapies.

Author response

  • Although we were not able to make major modifications in the abstract due to the policy of this journal (i.e. Abstract should be 200 words or less), we added the following rationale of comparison of Western and Eastern manual therapies, to the extent permitted.

“Background: Manual therapy (MT) is considered a promising adjuvant therapy for chronic obstructive pulmonary disease (COPD). Comparing the effectiveness among different Western and Eastern MTs being used for the management of COPD could potentially facilitate individualized management of COPD. This systematic review attempted to estimate the comparative effectiveness of Western and Eastern MTs for COPD patients using a network meta-analysis (NMA) methodology.” (Please see page 1, red words)

Comment 3

Introduction section:

In the introduction it is not clearly justified the need of comparison of Western and Eastern manual therapies.

Author response

  • We added the following rationale of comparison of Western and Eastern manual therapies

“Various manual therapies take a common approach when the practitioner's body comes in contact with the patient's body; therefore, they will have a common expected effect along with the unique effect of each therapy, which leads to unique results for multiple outcomes of COPD. For example, advocators of Western manual therapy may explain that by improving musculoskeletal changes of altered chest wall mechanics, manual therapy can affect chest wall compliance of patients with COPD [5]. On the other hand, Eastern manual therapies are regarded as manual therapy combined with traditional concept of meridian massage, and in this medical system, there is a view that both function and structure are systematically correlated [10]. Therefore, several Western and Eastern manual therapies can each have their own effectiveness for COPD, and comparative analysis of them can promote important individualized therapy in COPD management [11]. In addition, comparative analysis of Western and Eastern manual therapies for COPD could potentially help establish an integrative medical perspective for patients with COPD, by combining the advantages of each.” (Please see page 2, red words)

Comment 4

Methods section:

Defined inclusion criteria include: “Western and Eastern manual therapies”. A more exhaustive definition of the manual therapies included in each group should be provided, indicating the names of the manual therapies included in each group and the reasons.

Author response

  • Definitions and the criteria for Western and Eastern manual therapies have been added as follows:

“(3) Types of interventions: Western and Eastern manual therapies were included as interventions of interest, including manipulative therapy, joint mobilization, chiropractic, massage, reflexology, soft tissue therapy, muscle stretching, tuina, and acupressure passively applied using the practitioners’ hands. In this review, Western manual therapy was defined as manual therapy based on conventional Western anatomy. Specifically, manual therapy that mainly targets musculoskeletal changes of altered chest wall mechanics was considered Western manual therapy, which may include spinal manipulation, osteopathic manipulative treatment, manual diaphragm release technique, and soft tissue massage [5]. On the other hand, Eastern manual therapy was defined as manual therapy based on East Asian traditional medicine (EATM) theory such as meridian theory as well as conventional anatomy. Specifically, manual therapy targeting the meridian, a unique energy flow that connects the whole body in EATM, or based on a holistic perspective was considered Eastern manual therapy, which may include tuina, reflexology, and acupressure [10].” (Please see page 4, red words)

Comment 5

In Supplement 3. Results of pairwise meta-analysis.

  1. FEV1 (L): should be revised as data are not clearly view. Maybe range -20 to 20 should be modified.

Author response

  • Thank you for the careful advice. We revised the figures to increase readability.

  1. FEV1 (%pv): what this metanalysis add? There is not an overall result only the information provided study by study.

Author response

  • Thanks for your comments. Based on your comments, we present only the results of pairwise meta-analyses involving at least two studies in the revised manuscript and in Supplement 3.

Comment 6

The same occur in several of the next analysis. At least two studies should be included in each analysis. This does not occur for “4. FVC (%pv)”

Author response

  • Thanks for your comments. Based on your comments, we present only the results of pairwise meta-analyses involving at least two studies in the revised manuscript and in Supplement 3.

Comment 7

In the metanalysis Western and Eastern manual therapies are not compared.

Author response

  • Thanks for the comment. None of the included studies directly compared Western manual therapy with Eastern manual therapy. Therefore, we tried to indirectly compare individual manual therapies through network meta-analysis, and the results are described in the manuscript. Supplement 3 only describes the results of a pairwise meta-analysis of directly compared studies. Instead, we have strengthened the limitations of our review in response to your comments as follow.

“Finally, comparisons between manual therapies performed in this review primarily came from NMA, and the data are lacking in conventional pairwise meta-analysis. That is, the head-to-head trial comparing the comparative effects of different types of manual therapies for COPD patients in conventional RCTs was insufficient. In particular, head-to-head trials between Western manual therapy and Eastern manual therapy, one of the rationales of this systematic review, did not exist. Although the NMA methodology enables indirect comparison between interventions that have not previously been directly compared with each other [12], the overall poor methodological quality of the included studies suggests that large-scale, high-quality head-to-head trials can provide more reliable results. Given that various manual therapies are being used and studied for COPD patients in both East and West, robust clinical trials evaluating the comparative effectiveness of these treatments may be of interest to future researchers.” (Please see page 17, red words)